# The Impact of Neurocognitive Functioning on the Course of Posttraumatic Stress Symptoms following Civilian Traumatic Brain Injury

**DOI:** 10.3390/jcm10215109

**Published:** 2021-10-30

**Authors:** Dominique L. G. Van Praag, Filip Van Den Eede, Kristien Wouters, Lindsay Wilson, Andrew I. R. Maas

**Affiliations:** 1Department of Psychology, Antwerp University Hospital, 2650 Edegem, Belgium; 2Collaborative Antwerp Psychiatric Research Institute, University of Antwerp, 2610 Wilrijk, Belgium; 3Department of Psychiatry, Antwerp University Hospital, 2650 Edegem, Belgium; filip.vandeneede@uza.be; 4Clinical Trial Center, Clinical Research Center Antwerp, Antwerp University Hospital, 2650 Edegem, Belgium; kristien.wouters@uza.be; 5Antwerp Research Center, Faculty of Medicine, University of Antwerp, 2610 Wilrijk, Belgium; 6Division of Psychology, University of Stirling, Stirling FK9 4LA, UK; l.wilson@stir.ac.uk; 7Department of Neurosurgery, Antwerp University Hospital, 2650 Edegem, Belgium; andrew.maas@uza.be; 8Translational Neurosciences, University of Antwerp, 2610 Wilrijk, Belgium

**Keywords:** assessment, cognition, neuropsychology, posttraumatic stress, concussion, head injury

## Abstract

Background: One out of seven individuals who have suffered a traumatic brain injury (TBI) develops a posttraumatic stress disorder (PTSD), which is often associated with neurocognitive impairment. The present study explores the impact of neurocognitive functioning after mild, moderate, and severe TBI on the course of PTSD symptoms. Methods: The data of 671 adults admitted to hospital for a TBI was drawn from the Collaborative European Neurotrauma Effectiveness Research (CENTER-TBI) study. After six- and 12-months post-injury, participants completed the PTSD Checklist-5 (PCL-5), from which change scores were calculated. At six months, participants also completed a neurocognitive assessment including the Rey Auditory Verbal Learning Test, the Trail Making Test, and the Cambridge Neuropsychological Test Automated Battery (CANTAB). Linear regressions were performed to identify associations between cognitive functioning and PCL-5 change scores. Results: Overall, mean PCL-5 change scores showed no clear change (−0.20 ± 9.88), but 87 improved and 80 deteriorated by a change score of 10 or more. CANTAB Rapid Visual Information Processing scores were significantly associated with PCL-5 change scores. Conclusions: Strong sustained attention was associated with improvement in PTSD symptoms. Assessing cognitive performance may help identify individuals at risk of developing (persisting) PTSD post-TBI and offer opportunities for informing treatment strategies.

## 1. Introduction

Considering that 15% of patients suffering from mild traumatic brain injury (TBI), and 65% suffering from moderate or severe TBI, reported long-lasting cognitive problems and functional disability [1,2], it is evident that TBI is a chronic and incapacitating condition for many people. In addition to the cognitive, emotional, and physical consequences [3], in the case of civilian TBI, posttraumatic stress disorder (PTSD) is diagnosed in 14% [4,5]. TBI is a clear risk factor for PTSD, possibly through underlying neurobiological damage and impaired cognitive resources [6,7]. Poor cognitive performance may stem from the TBI but also, or alternatively, from post-TBI PTSD, since there is an overlap in the cognitive symptoms resulting from the two conditions, such as impaired speed of information processing, attention, working memory, learning and verbal memory [8,9,10]. Nonetheless, studies have reported differences in the cognitive deficits of patients with and without PTSD after mild TBI [11,12]. In a previous study by our research group, low cognitive flexibility and impaired long-term verbal memory were more strongly associated with probable PTSD than with TBI only [13]. Establishing differential cognitive profiles for co-occurring PTSD and TBI is crucial to informing treatment that targets cognitive impairment. Importantly, cognitive training after TBI should not merely focus on documented cognitive weaknesses but also on boosting (the remaining) protective cognitive resources. Activating the neurocognitive reserve may potentially protect against the development of PTSD in the longer term or help to reduce PTSD symptoms by enhancing the control over trauma memories and emotions while inhibiting maladaptive thoughts and producing constructive alternatives [14,15].

Vasterling et al. (2018) found that impairment of visual and verbal memory was related to an increase in PTSD severity over time [16]. They suggest that strong visual and verbal memory might help TBI survivors reconstruct their trauma narratives better and thus facilitate their processing of the traumatic event [17]. In contrast, Guo et al. (2017) report that in children, deficits in working memory and verbal learning three months post-TBI protected against PTSD at six months, while poor sustained attention predicted PTSD symptoms [18]. However, since these studies were conducted in military and paediatric populations, the results may not be generalizable to the broader context of adults having sustained TBIs in a civilian setting [16,18]. Additionally, the two studies covered either a relatively short or a long time span (up to 6 months versus an average of 7 years). As longitudinal research shows a peak in the prevalence of PTSD between six and 12 months after moderate to severe TBI, we considered it relevant to investigate symptom change within this timeframe [19].

The present longitudinal study builds on our previous work, in which we found low cognitive flexibility and impaired long-term verbal memory to be associated with probable PTSD in adults having suffered a civilian TBI [13]. Here, our primary aim is to describe the course of PTSD symptoms, more specifically to clarify the impact of neurocognitive functioning six months post-TBI on the course of PTSD symptoms between six and 12 months, where we hypothesise that strong visual and verbal memory will coincide with a reduction in PTSD symptoms. Our second aim is to identify the cognitive capacities that are key to an overall change in PTSD symptoms for each of the four PTSD clusters (i.e., intrusion, avoidance, cognition and mood, and arousal), which, to our knowledge, has not been previously studied in detail. In general, we expect that improved cognitive performance will be related to a decrease in PTSD symptoms [9,18,20]. We further anticipate that our exploratory analyses at the PTSD-cluster level will provide us with new insight into the relationship between cognition and PTSD-specific (intrusion) and non-specific symptoms (cognition and mood, and arousal). In addition to our main objectives, we will explore the evolution of PTSD by looking at the prevalence and recovery of ‘early’ PTSD at six months and the onset of ‘delayed’ PTSD 12 months post trauma.

## 2. Materials and Methods

### 2.1. Study Participants and Procedure

Participants with a clinical diagnosis of TBI were drawn from the European CENTER-TBI Core study (Collaborative European Neurotrauma Effectiveness Research: www.center-tbi.eu (accessed on 30 July 2021)), a prospective, observational study with 65 recruiting sites in 19 countries. Aimed at characterising TBI more closely and finding the most effective TBI treatment approaches (clinicaltrials.gov NCT02210221) [21], only participants who had presented at a study site within 24 h of the injury and who had had a clinical indication for CT-scan were included [22]. Pre-existing severe neurological disorders that could confound outcome assessments was an exclusion criterion. Participants were recruited in three strata: discharge from the accident and emergency room, admission to a hospital ward, or admission to the ICU. For the current study, we selected patients from the latter two strata only because, according to protocol, they were scheduled for the 12-month follow-up. Furthermore, participants needed to be over 16 years of age and have a 6-month Glasgow Outcome Score Extended (GOSE) [23] above 3, to thus exclude individuals who were likely to be unable to complete the cognitive assessment.

Research personnel collected demographic, pre-injury history, and injury-related information at recruitment. At the 6- and 12-month follow-up visits, self-report questionnaires were administered; at the 6-month follow-up, a trained research nurse or neuropsychologist administered the cognitive test battery. Research personnel were instructed to record test validity issues using test completion codes [24], and results flagged as invalid were removed. All data were entered into an electronic case report form, de-identified and stored in a secure database. Central or local Institutional Review Boards approved the protocol for all recruiting sites, and informed consent was obtained from all participants or their legally acceptable representative in accordance with national and local regulations. More information can be found on: https://www.center-tbi.eu/project/ethical-approval (accessed on 30 July 2021).

### 2.2. Measures

The Posttraumatic Stress Disorder Checklist, DSM-5 (PCL-5), is a self-report questionnaire to screen for PTSD, measure symptom severity and monitor change over time [25]. Respondents are asked to rate its 20 items that are based on the DSM-5 diagnostic criteria of PTSD on a scale ranging from 0 (not at all bothered) to 4 (extremely bothered), with a maximum total score of 80. Change scores can be calculated by subtracting the PCL-5 total score at six months from the PCL-5 total score at 12 months. Although change scores need to be determined and validated, a change of 10 points on the PCL-5 is considered clinically relevant, similar to the threshold for the PCL for the DSM-IV [25,26]. A probable diagnosis of PTSD can be made using the symptom cluster method that considers the DSM-5 cluster criteria [25,27], with items 1–5 reflecting intrusion symptoms, items 6 and 7 avoidance symptoms, items 8–14 cognition and mood, and items 15–20 arousal. A diagnosis requires at least a score of 2 or higher on one intrusion item, one avoidance item, two items on negative alterations in cognitions and mood and two items on alterations in arousal and reactivity. Cluster scores are calculated by adding these clinically relevant items. Below, we will alternately use the terms PTSD or probable or suspected PTSD when referring to the participants who met the PCL-5 criteria for PTSD. Note that to confirm the PCL-5 diagnosis, a thorough evaluation by a psychiatrist is required.

The Trail Making Test (TMT) is a pen-and-paper test assessing information processing, attention, and cognitive flexibility [28]. In Part A, the participant is instructed to connect numbers sequentially, as fast as possible, within 100 s and in Part B, numbers and letters alternately within 300 s.

The Rey Auditory Verbal Learning Test (RAVLT) assesses verbal learning and memory, and consists of a list of 15 unrelated words that are read to the participant, who is instructed to try and remember, and verbally reproduce as many words as possible [29,30]. This is repeated five times (trials 1–5: immediate recall), after which a second, so-called interference list is read out, which the participant needs to repeat once. Immediately after reproducing the interference words, the participant is asked to recollect words from the main list (trial 6: interference recall) and to recall the words again after 20 min (trial 7: delayed recall). The sum of the first five trials and trial six are used as a measure of short-term memory and the last trial as a measure of long-term memory.

The Cambridge Neuropsychological Test Automated Battery (CANTAB) is a computerised battery gauging diverse cognitive domains [31]. CANTAB mainly uses nonverbal stimuli in order to be language- and culture-independent. The reaction time task (RTI), the attention switching task (AST), the spatial working memory task (SWM), the paired associate learning task (PAL), the rapid visual information processing task (RVP) and the Stockings of Cambridge task (SOC) were selected for the present study. The RTI measures the intervals between the onset of the stimuli and the moments at which the participant releases the button, with the median duration serving as the outcome measure. The AST assesses cognitive flexibility by measuring the difference between response times for trial blocks in which the rule was switched and those for blocks in which the rule remained the same. Scores close to 0 indicate strong cognitive flexibility, since they reflect little variation in response times between switch and non-switch trials. The SWM assesses working memory by recording the number of errors made in finding hidden tokens relative to the maximum number of errors that could have been made. The PAL gauges visuospatial learning and memory. The number of errors is recorded and again compared to the maximum number of errors possible. Testing sustained attention and concentration, the RVP requires the participant to detect a specific sequence of digits; detection sensitivity is the outcome measure, with scores close to 1 reflecting strong sustained attention. Finally, the SOC tests for executive functioning, and more specifically for problem-solving strategies and spatial planning. The outcome is based on the number of trials completed in the smallest possible number of moves. See Appendix A for additional information on the TMT, RAVLT and CANTAB variables.

### 2.3. Statistical Analysis

SPSS version 25 was employed for all analyses [32]. CENTER-TBI data (version 2.1) were accessed using Neurobot, a bespoke data management tool (https://neurobot.incf.org (accessed on 1 July 2021)).

The cohort sample was described using the median and interquartile range (IQR) for continuous variables and the frequency and percentages for categorical variables. The PCL-5 change score was used to delineate the course of PTSD symptoms by subtracting the PCL-5 sum score at 6 months from the PCL-5 sum score at 12 months. In addition, we described the evolution of PTSD symptoms over time (from 6 to 12 months post trauma). Using the symptom cluster method, we identified four TBI groups: improved (suspected PTSD at 6 months only), delayed-onset PTSD (suspected PTSD at 12 months only), persistent PTSD (suspected PTSD both at 6 and 12 months) and resilient (no suspected PTSD at 6 or 12 months).

To determine the impact of neurocognitive functioning on the course of PTSD (cluster) symptoms, linear regressions were performed with the PCL-5 (cluster) change score as the dependent variable. Age, gender, educational level, psychiatric history, and the baseline Glasgow Coma Score (GCS) [33] were entered as demographic and injury-related covariates. The PCL-5 diagnosis at 6 months was entered to control for existing PTSD, as we expect the direction and sizes of change scores to be dependent on diagnostic status. The neurocognitive variables were TMT-A, TMT-B, RAVLT-Immediate recall, RAVLT-Interference recall, RAVLT-Delayed recall, CANTAB RTI, AST, PAL, SOC, SWM and RVP. All cognitive test scores were converted to z-scores for comparability across tests. In the case of non-normality, variables were first log-transformed. We explored the interaction effects of TBI severity and cognitive test scores on the PCL-5 change score. Model selection was based on covariate significance (*p* < 0.20) and adjusted R². Multicollinearity was checked by means of the variance inflation factor (VIF). Statistical significance was judged based on a more stringent *p*-value of <0.01 to decrease the risk of type-I errors.

## 3. Results

### 3.1. Study Sample and Characteristics

Of the 3661 patients of the CENTER-TBI Core study enrolled in the admission and ICU strata, 2072 met the inclusion criteria for this study. Of these, 671 participants had complete PCL-5 (6 + 12 months) and 6-month cognitive datasets and were included in the analyses (Figure 1).

Table 1 summarises the demographic and clinical characteristics of the study sample. The median age was 49, two thirds were male, and 70.6% was classified at baseline as having mild, 10% moderate and 19.4% severe TBI. One in ten had a psychiatric history, which mostly concerned a depressive disorder. At six months, 79 participants (11.8%) screened positive for PTSD and at 12 months, 71 participants screened positive (10.6%). In Appendix A, we list the characteristics of our sample and those of the participants (*n* = 1401) that met the inclusion criteria but had incomplete follow-up data. The participants lost to follow-up were older, had received less education and were more often single, living alone, or a combination thereof. In most participants of the study sample, TBI was sustained in traffic incidents, while in the lost-to-follow-up group, incidental falls were more frequent, with fewer suicide attempts or violent incidents being reported. Finally, in the lost-to-FU group, more people reported a history of substance abuse.

### 3.2. Course of PTSD Symptoms

The mean PCL-5 change score was −0.20, with a standard deviation of 9.88. A total of eighty-seven participants had improved (change score ≤ −10) and 80 had deteriorated (change score ≥ 10). The mean PCL-5 change score was significantly different (Hedges’ d = 1.03) for the participants meeting the criteria for probable PTSD at six months (M = −8.75, SD = 14.06) than it was for those that did not (M = 0.94, SD = 8.57), t(85.89) = 5.98, *p* < 0.001, Figure 2). Where the PTSD group showed a negative mean PCL-5 change score, reflecting an overall improvement of PTSD symptoms over time, the TBI-only group showed no overall statistically significant change. See Appendix A for the mean PCL-5 scores at 6 and 12 months. The mean PCL-5 cluster change scores are shown in Appendix A for the participants with and without probable PTSD.

The analysis investigating the evolution of PTSD symptoms over time revealed that of the 79 (of 671) participants with probable PTSD six months post-injury (11.8%), 40 (50.6%) had improved at 12 months. However, with 71 participants reporting symptoms at 12 months, PTSD prevalence had not significantly decreased relative to the 6-month assessment. In 39 of these 71 participants PTSD was persistent, while 32 had developed delayed-onset PTSD. The remaining 560 participants (83.5%) had no PTSD diagnosis at either timepoint (resilient group). Figure 3 shows the relationships between the PCL-5 scores at six months and the PCL-5 change scores differentiated for the four groups. Note that the maximum negative change score is dependent on the PCL-5 total score at six months, which, as can be seen in Figure 3, has a strong floor effect. In addition, Figure 3 shows regression to the mean of the PCL-5 scores. More than 80% of the resilient participants showed no clinically relevant change in PCL-5 scores, but in half of those with a persistent PTSD, PCL-5 scores had improved or worsened (>10 points, 1 SD). Conversely, those having improved or those with delayed-onset PTSD, showed no significant changes in PCL-5 scores (resp. 20 and 30%) and scored just above or close to the threshold at six months.

### 3.3. Cognitive Functioning and the Course of PTSD Symptoms

A multiple linear regression model was calculated to associate PCL-5 change scores with the cognitive test results (see Table 2, adjusted R² = 0.14). Age, psychiatric history, baseline GCS, probable PTSD diagnosis at six months, CANTAB RVP and AST were included in the final model. GCS was included as a continuous (ordinal) variable ranging from 3 to 15. Adding gender, educational level or other cognitive test outcomes had no added explanatory value. Baseline GCS was inversely related to changes in PTSD symptoms, with low GCS scores reflecting a severe injury and high PCL-5 change scores a worsening of PTSD symptoms (from 6 to 12 months). Suspicion of PTSD at six months was related to a reduction in PTSD symptoms, probably because of the room for improvement. Age and prior psychiatric illness showed no significant associations with PCL-5 change scores.

RVP outcomes were significantly inversely associated with PCL-5 change scores, with high RVP scores (strong sustained attention) relating to negative change scores, thus reflecting improved PTSD symptoms. The association between AST outcomes (where higher scores reflect poorer cognitive flexibility) and change scores was not significant (*p* = 0.014).

A subgroup analysis was performed for the 79 participants with probable PTSD at six months, to investigate the associations between PTSD symptom course and cognitive functioning. See Appendix A for the linear regression model (adjusted R² = 0.18). Higher CANTAB PAL scores (weak visual learning and memory) were significantly associated with higher change scores, reflecting a deterioration in PTSD symptoms. Low RAVLT-Interference recall scores (weak short-term verbal memory), but only in individuals with moderate and severe TBI, were significantly associated with higher PCL-5 change scores, reflecting improved PTSD symptoms.

### 3.4. Cognitive Functioning and the Course of PTSD Cluster Symptoms

The final models for the PTSD-cluster change scores are shown in Table 3. The change score for the intrusion and cognition and mood clusters were not significantly related to performance on any of the cognitive tests.

CANTAB AST performance was significantly related to avoidance change scores, but only in participants with moderate and severe TBI (see post-hoc test below Table 3, Appendix A). High AST scores (low cognitive flexibility) were related to high avoidance change scores, reflecting a worsening of avoidance symptoms. This also applied to RTI scores, but here, the interaction effect for the avoidance change score was not significant. High RTI scores (long reaction times) were related to high avoidance scores (worsening of avoidance symptoms).

CANTAB AST scores were significantly associated with arousal symptoms, but only in severe TBI (Appendix A). Higher AST scores (low cognitive flexibility) were related to deteriorating arousal symptoms. CANTAB RVP and RAVLT-Delayed recall scores were not significantly associated with arousal symptoms.

## 4. Discussion

Seeking to explore the effects of cognitive functioning six months post-TBI on the course of PTSD symptoms (as assessed with the PCL-5), we performed a complete case analysis on 671 participants with mild, moderate, or severe head injuries. Overall, nearly 25% (167/671) had PTSD symptom change scores of 10 or more points, with 87 participants having improved and 80 deteriorated. 11.8% had a probable diagnosis of PTSD at six months and showed significant improvement of their PTSD symptoms at 12 months compared to those without such a diagnosis, reflecting the difference in room for improvement. However, at the 12-month assessment, 32 new participants met criteria for delayed-onset PTSD.

Our hypothesis was only partially confirmed. We found a positive association between cognitive functioning and improvement of PTSD symptoms, however the strongest association was determined for sustained attention instead of verbal or visual memory, as hypothesized. Strong sustained attention (CANTAB RVP) was significantly related to improvement in PTSD symptoms after controlling for age, psychiatric history, baseline GCS and probable PTSD diagnosis. Looking at cognitive performance in relation to the four PTSD clusters, we found that none of the tests showed associations with the intrusion or cognition and mood clusters, while reaction time and cognitive flexibility correlated differentially with changes in the avoidance and arousal clusters.

### 4.1. Course of PTSD Symptoms

Since little research has been performed on the course of PTSD symptoms following TBI, comparisons with other findings are limited. We used a 10-point cut-off for the PCL-5 change score to indicate clinically relevant changes. Weathers and colleagues (2013) suggested a threshold of 5–10 for a reliable and 10–20 points for a clinically significant change [25]. This is in line with the distribution of the change scores in our study sample. Alway and Gould et al. (2016) found that 17.6% of their cohort developed PTSD within the first four years after sustaining moderate or severe TBI, with a peak between six and 12 months [19]. About two-thirds were categorised as showing delayed-onset PTSD (at least 6 months post trauma), of which one in three had subsyndromal symptoms. Even though our sample also included participants with mild TBI, we found similar results: 16.6% had probable PTSD six months or one-year post-injury, and one in three of the participants suffering from delayed-onset PTSD (12 months post-TBI) showed subsyndromal symptoms at six months. As far as we are aware, the only two previous studies that investigated the course of post-TBI PTSD symptoms in relation to cognition either focused on children or veterans and did not describe the most relevant period between six and 12 months [16,18].

### 4.2. Cognitive Functioning and the Course of PTSD Symptoms

Our results show an inverse relationship between CANTAB RVP (high scores reflect better sustained attention) scores and PTSD change scores six- and 12-months post-injury (negative scores reflect improving symptoms), reflecting that strong sustained attention led to a reduction in PTSD symptoms. Comparing sustained attention three months post-injury with PTSD symptoms at six months in children, Guo et al. (2017) found the same association [18]. Additionally, having a probable PTSD diagnosis six months post-TBI predicts improvement of symptoms up to 12 months; this likely reflects the fact that they have more opportunity to recover compared to participants without PTSD. Nonetheless, in both groups with and without probable PTSD, cognitive performance at six months could predict either an improvement or a deterioration in PTSD symptoms. As mentioned above, the severity of TBI is another predictor of symptom change, with Alway and colleagues [19,34] also reporting severe TBI to be associated with delayed-onset PTSD.

### 4.3. Cognitive Functioning and the Course of PTSD Cluster Symptoms

According to the DSM-5, the PTSD syndrome consists of four symptom clusters, of which the intrusion cluster is PTSD-specific. In the PCL-5, anxiety and depressive symptoms are captured in the cognition and mood, and arousal clusters, but they are not specific to PTSD [19,35]. Considering symptom changes in the four clusters, we found that none of the cognitive tests were significantly associated with change in the PTSD-specific intrusion cluster, which is similar to the findings by Guo et al. (2017) in a paediatric population [18]. They found a relationship between sustained attention scores at three months post injury and the non-PTSD-specific hyperarousal cluster at six months, which we could not replicate. In our sample, improvement in arousal symptoms was related to strong cognitive flexibility, but only in severe TBI. Additionally, we found that strong cognitive flexibility and reaction speed were related to an improvement in avoidance symptoms. Although this observation is a new finding for TBI, the relationship between cognitive flexibility and change in PTSD symptoms is a well-established association in populations with non-head trauma [36,37,38]. Ben-Zion et al. (2018) found that the cognitive training that improved cognitive flexibility also improved PTSD symptoms over time [36]. This suggests that interventions targeting cognitive flexibility positively affect PTSD development, support recovery, or a combination thereof, and may thus be a useful addition to traditional psychotherapeutic or pharmacological treatments [20].

### 4.4. Limitations

In the current study, we used the PCL-5 self-screening tool to monitor changes in PTSD severity but also to control for PTSD in our exploratory model. To arrive at this diagnosis of probable PTSD, we employed the symptom cluster method [24] but recognised that a formal diagnosis requires the structured assessment by a health professional. Further, the PCL-5 scores cannot be definitively related to the event that caused the TBI, as we have no pre-TBI data about PTSD symptoms. Additionally, change scores do not take the different symptom clusters into account and improvements can thus solely be attributable to changes in mood and not to recovery in PTSD-specific symptoms. To address this, we analysed changes for each of the four PTSD clusters separately. When interpreting cognitive test results, especially in patients with TBI, we need to acknowledge possible response bias due to lack of effort [39]. Although we did not include a formal performance validity test, examiners were instructed to record if low effort was apparent, and these test scores were removed from the database. Additionally, the TMT and RAVLT scores were examined and did not show evidence of systematic problems of low effort [40,41]. Further, to explore the impact of different cognitive variables on the course of PTSD, the choice of test is important. Great care was therefore taken in composing the test battery for the CENTER-TBI study; it needed to allow for cultural and language differences, since study participants were included from 19 European countries, and we needed to ensure that participants with mild, moderate and severe TBI would all be able to complete the full assessment. Still, both the TMT-B and CANTAB AST gauge cognitive flexibility, but we found only scores on the latter to be significantly associated with the observed changes. Of course, a cognitive test never addresses a single, isolated ability since cognitive functions are not separate constructs but overlap [42]. For example, cognitive flexibility also relies on attentional capacities and impaired sustained attention will affect learning and memory. Moreover, compared to the participants lost to follow-up, the participants in our study sample were younger, had received more education, were more often married or co-habiting and less frequently reported a history of substance abuse, while their TBI more often stemmed from road accidents, violent situations or a suicide attempt, whilst it more often stemmed from incidental falls in the lost-to-follow-up group. Furthermore, even though the cognitive covariates we computed were highly significant, we recognise that the adjusted R²s were low, indicating that their predictive value was limited. We also recognise that model selection may increase the risk of type-I errors, and we therefore used a more stringent significance level of *p* < 0.01. Finally, we did not control for the potential effects of psychopharmacological treatment or cognitive interventions, nor for life events that could have occurred after the TBI.

## 5. Conclusions

This study is the first to examine the course of self-reported PTSD symptoms (PCL-5) and associations with cognitive functioning in a relatively large civilian population with mild, moderate, and severe TBI. Overall, 13% of the participants showed improvement and 12% deterioration in PTSD symptoms from 6 to 12 months post-injury. The participants with suspected PTSD six months post-TBI (11.8%) showed an overall amelioration of PTSD symptoms compared to those that did not meet the PCL-5 criteria. Poor cognitive functioning has a negative impact on the course of PTSD symptoms following TBI. Assessment of reaction speed, sustained attention, and cognitive flexibility may accordingly help to identify patients at risk of developing (more severe) PTSD following TBI, with individual cognitive strengths and weaknesses informing treatment approaches. Future research should seek confirmation of our findings and, following recent reports [36,43], investigate whether cognitive training specifically aimed at enhancing reaction speed, sustained attention, and cognitive flexibility can alter the course of PTSD after TBI.

## Figures and Tables

**Figure 1 jcm-10-05109-f001:**
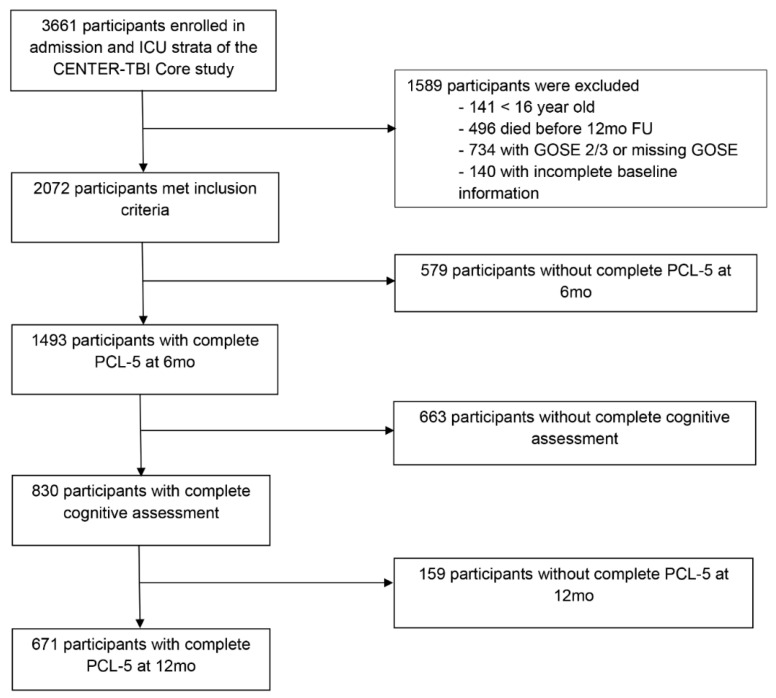
Participant inclusion flowchart.

**Figure 2 jcm-10-05109-f002:**
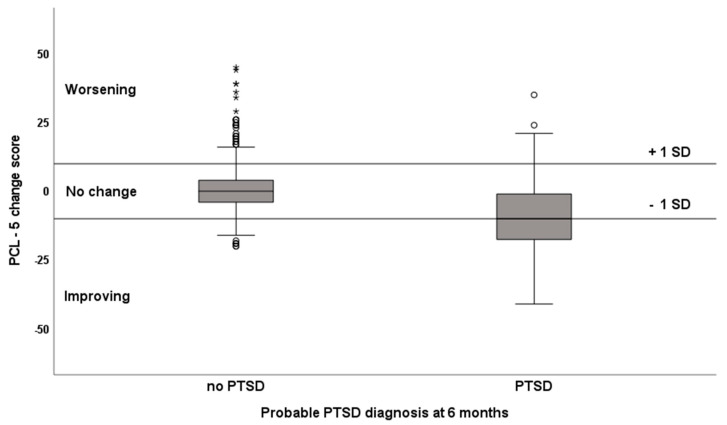
Boxplots of PCL-5 change scores differentiated for probable PTSD 6 months post-injury. The boxplots demonstrate the interquartile range and median of the PCL-5 change scores for participants with and without probable PTSD at 6 months. The whiskers represent 1.5× IQ range, ° are outliers between 1.5 and 3× IQ range, * are outliers over 3× IQ range.

**Figure 3 jcm-10-05109-f003:**
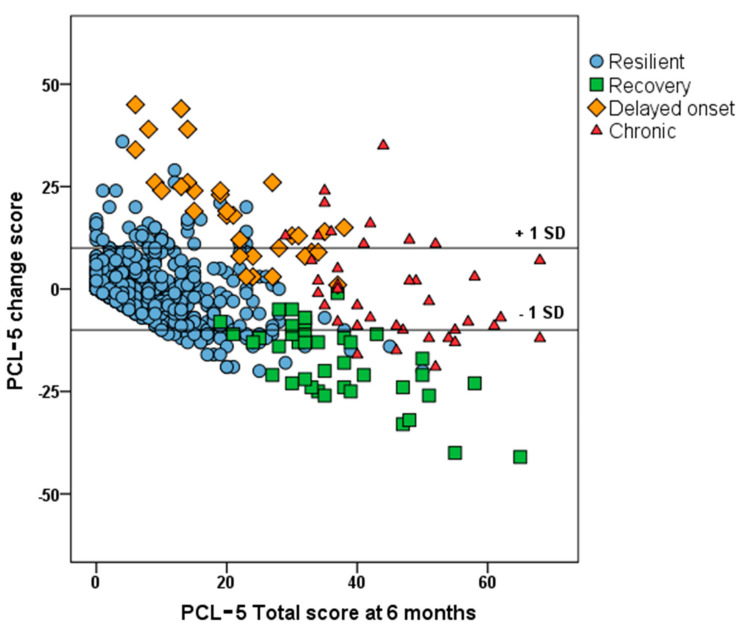
Scatterplot of PCL-5 change scores and PCL-5 total scores 6 months post-injury differentiated for PTSD at 6 and 12 months (resilient (83.5%), improved (5.9%), persistent (5.8%) and delayed onset (4.8%)).

**Table 1 jcm-10-05109-t001:** Participant characteristics (*n* = 671).

Demographic Characteristics	Median (IQR) ^1^ or *n* (%)
Age (years) >65 years	49 (31–61) 102/671 (15.2)
Male Highest educational level: College/University Married or living with partner	463/671 (69.0) 188/614 (30.6) 369/648 (56.9)
Injury-related characteristics	
Glasgow Coma Scale: Mild TBI (13–15) Moderate TBI (9–12) Severe TBI (3–8)	474/671 (70.6) 67/671 (10.0) 130/671 (19.4)
Cause of injury: Road traffic incident Incidental fall Violence/assault/act of mass violence Suicide attempt	321/656 (48.9) 244/656 (37.2) 24/656 (3.7) 12/656 (1.8)
Care pathway: Admitted to hospital Intensive Care Unit	32/671 (47.7) 351/671 (52.3)
Psychiatric history ²	
Psychiatric disorders Type of psychiatric disorder: Depression Anxiety Sleep disorder Substance abuse	68/671 (10.1) 37/68 (54.4) 18/68 (26.5) 11/68 (16.2) 8/68 (11.8)
Psychiatric characteristics at follow-up	
PCL-5 ^3^ total score 6 months PTSD probable diagnosis 6 months PCL-5 total score 12 months PTSD probable diagnosis 12 months PHQ-9 ^4^ total score 6 months GAD-7 ^5^ total score 6 months Psychotropic medication at 6 months Type of medication: Antidepressants Anxiolytics Antipsychotic agents	8 (3–18) 79/671 (11.8) 8 (2–18) 71/671 (10.6) 3 (1–7) 2 (0–5) 143/633 (22.6) 38/143 (26.6) 21/143 (14.7) 11/143 (7.7)

^1^ IQR = Interquartile range. ^2^ Information on psychiatric history and type(s) of psychiatric disorder was obtained during the interview with the patient and/or caretaker upon hospital admission. ^3^ PCL-5 is Posttraumatic Stress Disorder Checklist-5, ^4^ The PHQ-9 is the Patient Health Questionnaire-9, a self-report questionnaire that screens for depressive symptoms. It includes 9 items and results in a maximum total score of 27 with high scores reflecting more severe symptoms, ^5^ GAD-7 is Generalized Anxiety Disorder-7, a self-report questionnaire that screens for anxiety symptoms. It includes 7 items with a maximum total score of 21, with high scores reflecting more severe symptoms.

**Table 2 jcm-10-05109-t002:** Linear regression: covariates associated with PCL-5 change scores.

Covariate	B	SE (B)	95% CI ^1^		*p* ^4^	VIF ^5^
			LL ^2^	UL ^3^		
(Intercept)	−2.45	1.94	−6.25	1.35	0.21	
Age	−0.032	0.022	−0.075	0.012	0.15	1.22
Psychiatric history	2.14	1.18	−0.18	4.46	0.071	1.01
GCS ^6^	−0.27	0.095	−0.46	−0.082	0.005	1.23
PTSD diagnosis 6mo	−10.09	1.11	−12.28	−7.91	<0.001	1.02
CANTAB ^7^ RVP ^8^	−1.04	0.40	−1.82	−0.26	0.009	1.16
CANTAB AST ^9^	2.67	1.08	0.55	4.79	0.014	1.19
GCS × CANTAB AST	−0.21	0.086	−0.38	−0.037	0.017	

^1^ CI = confidence interval; ^2^ LL = lower limit; ^3^ UL = upper limit; ^4^ Significance level *p* < 0.01; ^5^ VIF = variance inflation factor; ^6^ GCS = Glasgow Coma Scale; ^7^ CANTAB = Cambridge Neuropsychological Test Automated Battery; ^8^ RVP = Rapid Visual Information Processing test (high scores reflect strong sustained ability); ^9^ AST = Attention Switching Task (high scores reflect low cognitive flexibility), high PCL-5 change scores reflect worsening symptoms. The R² change score for the CANTAB RVP is 0.008.

**Table 3 jcm-10-05109-t003:** Linear regression analyses: covariates associated with PCL-5 cluster change scores.

Covariates	Cluster Intrusion (5 Items)			Cluster Avoidance (2 Items)			Cluster Cognition/Mood (7 Items)			Cluster Arousal (6 Items)		
	B (SE)	*p* ^1^	VIF ^2^	B (SE)	*p* ^1^	VIF ^2^	B (SE)	*p* ^1^	VIF ^2^	B (SE)	*p* ^1^	VIF ^2^
Intercept	−0.98 (0.58)	0.092		−0.79 (0.26)	0.003		2.32 (0.84)	0.006		−0.84 (0.63)	0.18	
Age							−0.014 (0.010)	0.16	1.13			
Psychiatric history	0.84 (0.38)	0.027	1.01				0.25 (0.57)	0.66	1.00	0.53 (0.41)	0.20	1.01
GCS ^3^	−0.051 (0.029)	0.080	1.02	−0.029 (0.016)	0.060	1.03	−0.12 (0.046)	0.007	1.11	−0.092 (0.031)	0.003	1.03
PTSD 6mo	−2.64 (0.36)	<0.001	1.01	−1.26 (0.19)	<0.001	1.01				−2.67 (0.38)	<0.001	1.02
CANTAB ^4^ RVP ^5^	−0.23 (0.12)	0.053	1.03				−0.40 (0.19)	0.033	1.08	−0.31 (0.15)	0.031	1.29
CANTAB AST ^6^				0.70 (0.18)	<0.001	1.05				1.05 (0.38)	0.005	1.11
GCS × CANTAB AST				−0.050 (0.015)	0.001					−0.083 (0.030)	0.006	
CANTAB RTI ^7^				0.46 (0.17)	0.006	1.14						
GCS × CANTAB RTI				−0.034 (0.014)	0.013							
RAVLT-Delayed recall ^8^										0.31 (0.14)	0.023	1.21
F (df, between and within)	F(4, 666) = 16.10	<0.001		F(6, 664) = 12.68	<0.001		F(4, 666) = 19.05	<0.001		F(7, 663) = 11.07	<0.001	
Adjusted R²	0.083			0.095			0.020			0.095		

^1^ Significance level *p* < 0.01; ^2^ VIF = variance inflation factor; ^3^ GCS = Glasgow Coma Scale; ^4^ CANTAB = Cambridge Neuropsychological Test Automated Battery; ^5^ RVP = Rapid Visual Information Processing test (high scores reflect strong sustained ability); ^6^ AST = Attention Switching Task (high scores reflect low cognitive flexibility); ^7^ RTI = Reaction Time test (high scores reflect long reaction time); ^8^ RAVLT Delayed recall = Rey Auditory Verbal Learning Test—Delayed recall (high scores reflect strong long-term memory), high PCL-5 cluster change scores reflect worsening symptoms. The interaction effect for avoidance change scores and CANTAB AST, and for the arousal change scores and CANTAB AST were significant; (CANTAB AST in mild TBI: B = 0.003, SE(B) = 0.069, *p* = 0.97, moderate TBI; B = 0.18, SE(B) = 0.064, *p* = 0.006 and severe TBI: B = 0.43, SE(B) = 0.11, *p* < 0.001) and (CANTAB AST in mild TBI: B = −0.11, SE(B) = 0.15, *p* = 0.45, moderate TBI: B = 0.18, SE(B) = 0.13, *p* = 0.18 and severe TBI: B = 0.60, SE(B) = 0.23, *p* = 0.009), respectively. The R² change score for the CANTAB AST and the interaction between CANTAB AST and GCS for the avoidance change scores is 0.018. The R² change score for the CANTAB RTI and the interaction between CANTAB RTI and GCS for the avoidance change scores is 0.008. The R² change score for the CANTAB AST and the interaction between AST and GCS for the arousal change scores is 0.008.

## Data Availability

CENTER-TBI is committed to data sharing and in particular to responsible further use of the data. Hereto, we have a data sharing statement in place: https://www.center-tbi.eu/data/sharing/ (accessed on 30 July 2021). The CENTER-TBI dataset is hugely complex and the CENTER researchers wish to encourage correct and appropriate use of the data; this means that we encourage researchers to contact the CENTER-TBI team for any research plans and the Data Curation Team for any help in the appropriate use of the data, including the sharing of scripts. Requests for data access can be submitted online: http://www.center-tbi.eu/data/ (accessed on 30 July 2021). The complete Manual for data access is also available online: https://www.center-tbi.eu/files/SOP-Manual-DAPR-20181101.pdf/ (accessed on 30 July 2021).

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
