# Peer review of "The Impact of Neurocognitive Functioning on the Course of Posttraumatic Stress Symptoms following Civilian Traumatic Brain Injury"

_jcm, 2021, doi:10.3390/jcm10215109_

Round 1

Reviewer 1 Report

This is a very well-written paper on an impressive sample; however, I have several concerns with the current manuscript, which I believe preclude publication in such an esteemed journal.

My main concerns with this study include the following:

-Responses on the PCL-5 were not tied to the injury event. If a person experienced a traumatic event in childhood, for instance, that could have been driving their responses. Similarly, it is possible that individuals with delayed onset were responding to a traumatic event that occurred after the 6 month evaluation.

-No consideration of performance validity, which is essential in studies of TBI involving cognition (could it be that those with the worst effort also had worsening reported PTSD symptoms over time and that is what is making it seem like better cognition leads to improvements in PTSD symptoms)?

-no consideration of symptom validity

-It seems to me a threshold of p<.01 is not adequate. For Table 2, the authors investigated 11 cognitive tests, as well as their interaction with GCS. That is 22 different analyses of interest.  For Table 3, the authors investigated these same 11 cognitive tests and their interaction with GCS across 4 different dependent variables, for a total of 88 analyses of interest.

Other considerations:

-How was baseline GCS entered into the model? Continuous? 2 groups? 3 groups?

-Only 68 individuals were considered to have a psychiatric history, yet 143 were on psychotropic medications. Why this discrepancy?

-It seems less likely that cognition impacts PTSD and more likely that PTSD impacts cognition, but the data you have better fits investigation of the former.

-Might regression to the mean be playing a role here?

-In Table 2, the AST is not p<.01, nor is the interaction between GCS and AST. Similarly, there are some cognitive variables in Table 3 that are not p<.01.

-It would be nice to see the R squared change after adding in the variable of interest, not just the adjusted R squared for the entire model.

-I am not sure the change score is as important as the total score. When using change scores, an individual who is asymptomatic on the PCL at baseline and remains asymptomatic looks the same as an individual who reports severe symptoms on the PCL and continues to do so; however, these individuals are very different.

-Much of the literature on PTSD and TBI has occurred within the context of mild TBI samples. In the introduction, when data comes from mild TBI only vs. moderate/severe TBI only, this should be clarified. It can be misleading to imply these relationships have been demonstrated across the TBI severity spectrum.

Author Response

Response to reviewer #1

This is a very well-written paper on an impressive sample; however, I have several concerns with the current manuscript, which I believe preclude publication in such an esteemed journal.

My main concerns with this study include the following:

  • Responses on the PCL-5 were not tied to the injury event. If a person experienced a traumatic event in childhood, for instance, that could have been driving their responses. Similarly, it is possible that individuals with delayed onset were responding to a traumatic event that occurred after the 6 month evaluation.

We agree with the reviewer that this is a limitation of our study. This is a cross-sectional study as regards to the PTSD symptoms and the cognitive functions. For an acute event, such as TBI, in the general population, it is impossible to obtain information on baseline or pre-TBI-data about PTSD symptoms. This stands in contrast to studies in, for example, military or selected sports populations, in which baseline values can be established. We added the following sentence to the limitations section (lines 398-391): “Further, the PCL-5 scores cannot be definitively related to the event that caused the TBI, as we have no pre-TBI-data about PTSD symptoms.”

Although we cannot definitively relate these symptoms to the (event that caused the) TBI six months before, we consider this unlikely. The follow-up interviews conducted at 6 and 12 months post-TBI were extensive, and detailed information on medical interventions as also traumatic events that may have occurred after the index event were captured. No patients with a new-onset PTSD reported a second event.

We further note that our study focused on change in PCL-5 scores between 6 and 12 months after injury, rather than on the score at 6 months. From this perspective, we consider it unlikely that a change (improvement) in PCL-5 scores between 6 and 12 months would occur if PTSD had been caused by a previous stressful event. New-onset PTSD at 12 months after injury might potentially have been caused by a second event (after discharge) in some cases, but our detailed follow-up interviews never documented such occurrence.

  • No consideration of performance validity, which is essential in studies of TBI involving cognition (could it be that those with the worst effort also had worsening reported PTSD symptoms over time and that is what is making it seem like better cognition leads to improvements in PTSD symptoms)?
  • No consideration of symptom validity

We thank the reviewer for his/her comment. CENTER-TBI is an observational study and we did not include a formal performance validity test (PVT)/ symptom validity test (SVT). The issue of possible invalid cognitive data is indeed a concern for neuropsychological testing. Aiming to reduce this risk, examiners were instructed to record if low effort or uncooperative behavior was apparent in testing for each cognitive test separately by test completion codes (Bagiella et al., 2010). The following was added in the Procedure paragraph (lines 205-207): “Research personnel were instructed to record test validity issues using test completion codes (Bagiella et al., 2010), and results flagged as invalid were removed.” One may argue, however, that instructing staff to report low effort is not a valid method for determining validity and test completion codes were not intended as a substitute for formal validity testing. We therefore looked at embedded measures of performance validity using the Trail Making Test A and B, as also the RAVLT.

Stenberg et al. (2020) used absolute cut-off scores for low performance on the RAVLT based on previous work by Boone at colleagues (2005): Trial 5 ≤ 6 and Trials1–5 ≤ 28. In our sample, 5.7% met the first criterion and 7.8% met the second criterion. This is similar to estimates of base rates of PVT failure for medical cases (Mittenberg et al., 2002) and much lower than rates of 39%-46% reported for cases with PTSD/ mTBI in disability exams (Young et al., 2016). Similar percentages were found when using the TMT as a measure for low performance. A paper by Iverson et al. (2002) presents three cut-off scores; TMT A time > 62, TMT B > 199, TMT B/A ratio ≤ 1.49. In our sample, 7.5%, 4.1% and 7% met the first, second and third criterion, respectively. Since the study included individuals with severe TBI, a proportion are expected to show low or very low performance on testing. This post-hoc analysis of performance on the RAVLT and the TMT thus suggested little in the way of systematic issues with validity. Given the relatively low rates of poor performance and the uncertain interpretation, we did not exclude individuals based on TMT or RAVLT performance. Given that there were no incentives to deliberately exaggerate reported symptoms we did not separately examine this issue. We added the following to the limitations section (lines 394-399): “When interpreting cognitive test results, especially in patients with TBI, we need to acknowledge possible response bias due to lack of effort (Wisdom et al., 2014). Although we did not include a formal performance validity test, examiners were instructed to record if low effort was apparent and these test scores were removed from the database. Additionally, the TMT and RAVLT scores were examined and did not show evidence of systematic problems of low effort (Boone et al., 2005; Iverson et al., 2002).”

  • It seems to me a threshold of p<.01 is not adequate. For Table 2 the authors investigated 11 cognitive tests, as well as their interaction with GCS. That is 22 different analyses of interest. For Table 3, the authors investigated these same 11 cognitive tests and their interaction with GCS across 4 different dependent variables, for a total of 88 analysis of interest.

We thank the reviewer for his/her comment and would like to explain our rationale for choosing a p-value of 0.01. We acknowledge we have a large number of variables, however a full correction for multiple testing using Bonferroni, Holm, FDR or similar, would fundamentally increase the probability of a type II error. As the nature of our study is exploratory, we searched for a balance between controlling type I error without missing possibly interesting associations and decided to adopt a pragmatic approach, specifying a p-value of 0.01. We have mentioned the following in the limitations section (in the first submitted version of the manuscript): “We also recognize that model selection may increase the risk of type-I errors, we therefor used a more stringent significance level of p < 0.01.”

Other considerations:

  • How was baseline GCS entered into the model? Continuous? 2 groups? 3 groups?

Baseline GCS was entered as a continuous (ordinal) variable ranging from 3 to 15. We added this (lines 264-265): “GCS was included as a continuous (ordinal) variable ranging from 3 to 15.”

  • Only 68 individuals were considered to have a psychiatric history, yet 143 were on psychotropic medications. Why this discrepancy?

At baseline, 68 reported a history of a psychiatric disorder. At 6 month follow-up, 143 individuals used psychotropic medications, including antidepressants, anxiolytics, antipsychotic agents, psychostimulants and other psychotropic medication. The discrepancy between the number of psychiatric history and the use of psychotropic medication at 6 months reflects the increase in psychiatric symptoms following TBI.

  • It seems less likely that cognition impacts PTSD and more likely that PTSD impacts cognition, but the data you have better fits investigation of the former.

We thank the reviewer for this interesting remark and believe that it is quite possible that there is a bi-directional effect. As the reviewer suggests, the development of PTSD (symptoms) does not solely rely on cognitive (dis)functioning and individuals suffering from PTSD can experience a range of cognitive difficulties. However, cognitive strengths or weaknesses might also facilitate recovery from PTSD. As described in the introduction, Vasterling and colleagues (2018) found that impairment of memory was related to greater PTSD severity and suggested that strong memory might help TBI survivors reconstruct their trauma narratives better and thus facilitate their processing of the traumatic event. The present study cannot conclude on a causal relation between cognition and PTSD symptom course, but aims to broaden the role of cognition from ‘result of PTSD’ to a possible target for treatment.

  • Might regression to the mean be playing a role here?

As the reviewer notices, the PCL-5 scores in Figure 3 show regression to the mean, however we do not expect this to have a (large) impact on the results. We added this to the text (lines 248-249): “In addition, Figure 3 shows regression to the mean of the PCL-5 scores.”

  • In Table 2, the AST is not p<.01, nor is the interaction between GCS and AST. Similarly, there are some cognitive variables in Table 3 that are not p<.01.

That is correct, model selection was, amongst other criteria, based on covariate significance using p<.20, as mentioned in section 2.3 of the manuscript.

  • It would be nice to see R² change after adding in the variable of interest, not just the adjusted R² for the entire model.

The R² change scores were added below Table 2: “The R² change score for the CANTAB RVP is .008.” and Table 3: “The R² change score for the CANTAB AST and the interaction between CANTAB AST and GCS for the avoidance change scores is .018. The R² change score for the CANTAB RTI and the interaction between CANTAB RTI and GCS for the avoidance change scores is .008. The R² change score for the CANTAB AST and the interaction between AST and GCS for the arousal change scores is .008.” The variables of interest for the cluster change scores were reported with their respective interaction effect variables, as we would not include the interaction effect without the main effect in the model.

  • I am not sure the change score is as important as the total score. When using change scores, an individual who is asymptomatic on the PCL at baseline and remains asymptomatic looks the same as an individual who reports severe symptoms on the PCL and continues to do so; however, these individuals are very different.

We agree with the reviewer that the PCL-5 total score is an important outcome following TBI. In a previous study (in review), we performed a cross-sectional study, exploring the relation between cognition and the PCL-5 total score and found that cognitive flexibility and verbal memory were significantly associated to the PCL-5 score. In the present study, we were interested to investigate which cognitive test scores were related to the change in PTSD symptoms. The difference between individuals without symptoms (resilient) or with symptoms (persistent) is out of scope. We realise that the maximum negative change score is dependent on the PCL-5 at six months. The floor effect was described in the results section and is shown in Figure 3.

  • Much of the literature on PTSD and TBI has occurred within the context of mild TBI samples. In the introduction, when data comes from mild TBI only vs. moderate/severe TBI only, this should be clarified. It can be misleading to imply these relationships have been demonstrated across the TBI severity spectrum.

Done. “mild TBI” was added in the introduction (line 47): “Nonetheless, studies do report differences in the cognitive deficits of patients with and without PTSD after mild TBI”. Other references of ‘TBI’ discuss TBI in general (mild to severe).

References

Bagiella, E.; Novack, T.A.; Ansel, B.; Diaz-Arrastia, R.; Dikmen, S.; Hart, T.; Temkin, N. Measuring outcome in traumatic brain injury treatment trials: recommendations from the traumatic brain injury clinical trials network. J Head Trauma Rehabil. 2010, 25(5), 375–82. https://doi.org/10.1097/HTR.0b013e3181d27fe3

Boone, K.B.; Lu, P.; Wen, J. Comparison of various RAVLT scores in the detection of noncredible memory performance. Arch Clin Neuropsychol. 2015, 20(3),301–19. https://doi.org/10.1016/j.acn.2004.08.001

Iverson, G.L.; Lange, R.T.; Green, P.; Franzen, M.D. Detecting exaggeration and malingering with the Trail Making Test. Clin Neuropsychol. 2002, 16, 398–406. doi:10.1076=clin.16.3.398.13861

Mittenberg, W.; Patton, C.; Canyock, E.M.; Condit, DC. Base rates of malingering and symptom exaggeration. J Clin Exp Neuropsychol. 2002, 24(8),1094–1102. https://doi.org/10.1076/jcen.24.8.1094.8379

Stenberg, J.; Karr, J.E.; Terry, D.P.; Saksvik, S.B.; Vik, A.; Skandsen, T.; Silverberg, N.D.; Iverson, G.L. Developing Cognition Endpoints for the CENTER-TBI Neuropsychological Test Battery. Front Neurol. 2020, 11, 670. https://doi.org/10.3389/fneur.2020.00670

Vasterling, J.J.; Aslan, M.; Lee, L.O.; Proctor, S.P.; Ko, J.; Jacob, S.; Concato, J. Longitudinal associations among Posttraumatic stress disorder symptoms, traumatic brain injury and neurocognitive functioning in Army Soldiers deployed to the Iraq War. J Int Neuropsychol Soc, 2018, 24(4), 311-323. https://doi.org/10.1017/S1355617717001059

Wisdom, N.M.; Pastorek, N.J.; Miller, B.I.; Booth, J.E.; Romesser, J.M.; Linck, J.F.; Sim, A.H. PTSD and cognitive functioning: importance of including performance validity testing. Clin Neuropsychol2014, 28(1), 128–145. https://doi.org/10.1080/13854046.2013.863977

Young, J.C.; Roper, B.L.; Arentsen, T.J. Validity testing and neuropsychology practice in the VA healthcare system: results from recent practitioner survey. Clin Neuropsychol. 2016, 30(4), 497–514. https://doi.org/10.1080/13854046.2016.1159730

Reviewer 2 Report

The present study was aimed at explores the impact of neurocognitive functioning after mild, moderate 20 and severe TBI on the course of PTSD symptoms at 6 and 12 months after the event.

Probable PTSD diagnosis was present in 11.8% and 10.6% of the sample at 6 and 12 months respectively. CANTAB Rapid Visual Information Processing scores were significantly associated with PCL-5 change scores. Additionally, strong cognitive flexibility and reaction speed were related to improvement in avoidance symptoms

It is a very interesting study that presents the merit of investigating prospectively PTSD symptoms after an event. Furthermore, objective measures of cognitive impairment were examined. The study methodology was clearly described and the study was well designed. Limitations were correctly reported. I have only minor comments to the Authors:

  • The traumatic event types: different kinds of traumatic events could have a different impact on PTSD symptoms onset. In particular, Violence and Suicide attempt are usually considered more associated with PTSD than accidents. In my opinion the authors could include also this variable in the regression analyses.

  • A brief description of the GAD-7 and PHQ-9 should be included. In my opinion they are not necessary for the aim of the study and, in fact, they were not utilized in the analyses. They could be used in the regression analysis as independent variables predicting PTSD at 12 months or eliminated from the results.

  • How previous psychiatric history was examined? Was a structured clinical interview used?

  • You should report percentage of the different groups based on PTSD presence at 6 and 12 months. In my opinion “chronic PTSD” and “recovered PTSD” groups could be reformulated. Recovery is a complex issue that goes beyond reaching diagnosis threshold. Furthermore, “persistent” could be more accurate than “chronic”.

Author Response

Reviewer #2

The present study was aimed at explores the impact of neurocognitive functioning after mild, moderate and severe TBI on the course of PTSD symptoms at 6 and 12 months after the event. Probable PTSD diagnosis was present in 11.8% and 10.6% of the sample at 6 and 12 months respectively. CANTAB Rapid Visual Information Processing scores were significantly associated with PCL-5 change scores. Additionally, strong cognitive flexibility and reaction speed were related to improvement in avoidance symptoms. It is a very interesting study that presents the merit of investigating prospectively PTSD symptoms after an event. Furthermore, objective measures of cognitive impairment were examined. The study methodology was clearly described and the study was well designed. Limitations were correctly reported. I have only minor comments to the authors:

  • The traumatic event types: different kinds of traumatic events could have a different impact on PTSD symptoms onset. In particular, violence and suicide attempt are usually considered more associated with PTSD than accidents. In my opinion the authors could include also this variable in the regression analyses.

We agree with the reviewer that the type of traumatic event is an important risk factor for PTSD. However, when investigating the change in PTSD symptoms, the type of traumatic event was not significantly related. We might conclude that the type of traumatic event has no impact on the improvement or worsening of PTSD symptoms after TBI. We recognize that the number of patients with a TBI resulting from a violent event or related to suicide, is relatively low (n=36). However, the boxplot below shows no significant difference in PCL-5 change scores for the different types of traumatic event.

  • A brief description of the GAD-7 and PHQ-9 should be included. In my opinion they are not necessary for the aim of the study and, in fact, they were not utilized in the analyses. They could be used in the regression analysis as independent variables predicting PTSD at 12 months or eliminated from the results.

We thank the reviewer for his/her suggestion and included a brief description of the GAD-7 and the PHQ-9 below the table: “The PHQ-9 is the Patient Health Questionnaire-9, a self-report questionnaire that screens for depressive symptoms. It includes 9 items and results in a maximum total score of 27 with high scores reflecting more severe symptoms. 5 GAD-7 is Generalized Anxiety Disorder-7, a self-report questionnaire that screens for anxiety symptoms. It includes 7 items with a maximum total score of 21, with high scores reflecting more severe symptoms.”. We agree that the measures have no added value in the final model, exploring the relation between cognition and PTSD symptom change following TBI. Information on the PHQ-9 and GAD-7 was therefore not added to the method section.

  • How previous psychiatric history was examined? Was a structured clinical interview used?

Information on the history of psychiatric disorders was based on information obtained from patient and/or carer upon presentation. We did not use a structured clinical interview. The structured approach in the e-CRF specified the following categories: Anxiety, Depression, Sleep disorder, Schizophrenia, Substance abuse disorder, and other.

  • You should report percentage of the different groups based on PTSD presence at 6 and 12 months. In my opinion, “chronic PTSD” and “recovered PTSD” groups could be reformulated. Recovery is a complex issue that goes beyond reaching diagnosis threshold. Furthermore, “persistent” could be more accurate than “chronic”.

We thank the reviewer for his/her suggestion. The percentages of the PTSD groups were added to the Figure: “Scatterplot of PCL-5 change scores and PCL-5 total scores 6 months post-injury differentiated for PTSD at 6 and 12 months (Resilient (83.5%), Improved (5.9%), Persistent (5.8%) and Delayed onset (4.8%)).” As suggested, we reformulated ‘chronic’ to ‘persistent’, and ‘recovered’ to ‘improved’ group. Changes to the text were made accordingly in the statistical analysis section and in the text describing Figure 3 (lines 175-177, 241, 243, 251, 252, 257-259).
